# Chemotherapy Resistance Explained through Endoplasmic Reticulum Stress-Dependent Signaling

**DOI:** 10.3390/cancers11030338

**Published:** 2019-03-08

**Authors:** Entaz Bahar, Ji-Ye Kim, Hyonok Yoon

**Affiliations:** 1College of Pharmacy, Research Institute of Pharmaceutical Sciences, Gyeongsang National University, Jinju 52828, Gyeongnam, Korea; entaz_bahar@yahoo.com; 2Department of Pathology, College of Medicine, Yonsei University, Seoul 03722, Korea; 3Department of Pathology, Ilsan Paik Hospital, Inje University, Goyang 10381, Gyeonggi-do, Korea; 4Department of Pathology, National Cancer Center, Goyang 10408, Gyeonggi-do, Korea

**Keywords:** cancer, chemotherapy resistance, endoplasmic reticulum, endoplasmic reticulum stress response

## Abstract

Cancers cells have the ability to develop chemotherapy resistance, which is a persistent problem during cancer treatment. Chemotherapy resistance develops through different molecular mechanisms, which lead to modification of the cancer cells signals needed for cellular proliferation or for stimulating an immune response. The endoplasmic reticulum (ER) is an important organelle involved in protein quality control, by promoting the correct folding of protein and ER-mediated degradation of unfolded or misfolded protein, namely, ER-associated degradation. Disturbances of the normal ER functions causes an accumulation of unfolded or misfolded proteins in the ER lumen, resulting in a condition called “ER stress (ERS).” ERS triggers the unfolded protein response (UPR)—also called the ERS response (ERSR)—to restore homeostasis or activate cell death. Although the ERSR is one emerging potential target for chemotherapeutics to treat cancer, it is also critical for chemotherapeutics resistance, as well. However, the detailed molecular mechanism of the relationship between the ERSR and tumor survival or drug resistance remains to be fully understood. In this review, we aim to describe the most vital molecular mechanism of the relationship between the ERSR and chemotherapy resistance. Moreover, the review also discusses the molecular mechanism of ER stress-mediated apoptosis on cancer treatments.

## 1. Introduction

Cancer is one of the leading causes of death, worldwide, for instance there were about 15 million diagnosed cancer cases and 8.2 million deaths in 2013 [1]. Cancer is the second leading cause of death for amongst the United States population, after heart disease, and the leading cause of death for non-Hispanic, Asian, or Pacific Islander, and Hispanic populations [2]. According to the Korea National Statistical Office, cancer (malignant neoplasms) was significantly higher than deaths caused by cerebrovascular disease, heart disease, diabetes, suicide, and other deaths [3]. Like getting rid of weeds, healing from cancer becomes more and more difficult, day by day. The treatments that kill cancer cells are generally toxic to normal cells, as well [4]. The main objective of cancer treatment is to destroy cancer cells, while causing minimal damage to normal tissue, which can be achieved, either directly or indirectly, by modifying the signals needed for cellular proliferation in cancer cells or by stimulating an immune response [5,6]. The therapeutic management of cancer depends on the cancer type, its location and extent, the patient age, and other characteristics, including specific pathological, molecular, genetic, epigenetic, and microenvironmental changes in which the cancer cell resides [7,8,9,10]. Cancers can be treated with a combination of therapies (surgery, radiation, chemotherapy, laser therapy, and targeted therapy), chosen on the basis of the type and stage of cancer [11]. Cancer cells have the ability to develop resistance to chemotherapeutics, which is a persistent problem during cancer treatment [6,12]. Chemotherapy becomes resistant through different mechanisms, including patho-physiological, micro-environmental, genetic, and epigenetic changes in the tumor cell [13]. The increasing prevalence of chemotherapy resistance requires the development of further treatments and effective research.

The endoplasmic reticulum (ER) has multifunctional activities, including protein folding, protein maturation, and ER quality control (ERQC), to maintain a cellular homeostasis [14]. The perturbation of the normal ERQC system causes an accumulation of unfolded or misfolded proteins in the ER lumen, resulting in a condition called “ER stress (ERS)” [15]. Upon ERS, endoplasmic reticulum stress response (ERSR) is produced to restore homeostasis or activate cell death [16]. Several studies have suggested that the ERSR could be the potential target for chemotherapeutics to treat cancer [17,18]. Recently, it has been reported that the ERS is critical for chemo-therapeutics resistance, following the initiation of an ERSR [19,20,21,22]. Although a thorough understanding of the ERSR associated with cancer drug resistance will enable us to develop more effective chemotherapeutic candidates, the relationship between the ERSR and chemotherapy resistance remains poorly understood. In this review, we focused on the detailed molecular mechanism of the relationship between ER stress and tumor survival or drug resistance. In addition, the effects of ER stress-mediated apoptosis on cancer treatments are also presented.

## 2. Apoptosis as a Therapeutic Target for Anticancer Therapy

Chemotherapy, radiation therapy, and surgery are the main procedures associated with cancer management. The main objective of cancer therapy is to destroy all the cancer cells, while causing minimal damage to the normal tissue. Apoptosis or the process of programmed cell death is a genetically regulated form of cell death, and is an emerging target for anticancer therapy [23]. In recent years, the most vital advances in clinical oncology involve the killing of tumor cells, mostly by apoptosis, which crucially determines the treatment response. For example, current cancer therapies including chemotherapy, radiotherapy, suicide gene therapy, or immunotherapy, exhibit antitumor effects by activating the apoptosis signal transduction pathways in cancer cells [24]. There are three different pathways that lead to apoptosis, which are (a) the extrinsic death receptor pathway, (b) the intrinsic mitochondrial pathway, and (c) the ERS pathway (Figure 1). These pathways are activated by both intracellular and extracellular signals and converge at the executioner caspases, to carry out apoptosis through the cleavage of hundreds of proteins [25].

### 2.1. Extrinsic Death Receptor Pathway

The extrinsic pathway (death-receptor-dependent), belongs to the superfamily of tumor necrosis factor receptors (TNFR), while their respective protein TNF family ligands activate caspases, via cell-surface death receptors, which respond to the cognate death ligands expressed on immune-effector cells [26,27]. The extrinsic apoptosis pathway is an attractive target to trigger apoptosis in tumor cells for cancer therapy, as death receptors have been directly linked with the cell’s death machinery [28].

### 2.2. Intrinsic Mitochondrial Pathways

The intrinsic or death receptor apoptotic pathway activates caspases via members of the B-cell lymphoma-2 (Bcl-2) protein family and the mitochondria, as a reaction to severe cellular damage or stress [26]. Mitochondria play a vital role in cellular processes, such as cellular energy metabolism, cell differentiation, and regulation of the cell cycle. In addition, mitochondria control apoptosis through the regulated suppression or activation of pro-apoptotic proteins, electron transport control, and maintenance of cellular redox potential [29,30]. Mitochondria are targets with great potential as therapeutics for cancer. Mitochondrial dysfunction activates apoptotic processes that lead to changes in the mitochondrial structure, loss of electron transport function, and disruption of the mitochondrial membrane potential, leading to the release of reactive oxygen species (ROS), caspase activation, and activation of other primary mediators of apoptosis [31,32].

### 2.3. Endoplasmic Reticulum Stress Pathways

ER is an important organelle involved in multiple cellular activities, including synthesis, maturation, translation, and folding of secretory and membrane proteins, lipid biogenesis, and the sequestration of Ca^2+^ [33,34,35]. Various physiological and pathological conditions might affect ER homeostasis, causing ERS. The homeostatic mechanisms that maintain the balance of large numbers of proteins are ultimate targets of novel anticancer strategies [36]. Actually, ER signaling pathways are an emerging target for cancer therapeutics. ERS is of great interest to cancer biologists, as both the intrinsic mitochondrial and ER stress-mediated apoptosis pathways are activated by each other’s mechanisms.

## 3. Endoplasmic Reticulum Stress-Mediated Apoptosis and Cancer Therapy

ERS is used as an emerging therapeutic target for cancer treatment. Recently, a number of major hallmarks of cancer have been identified that are expected to facilitate the development of anticancer therapies. For example, a drug combination of rapamycin and the heat shock protein 90 (HSP90) inhibitor retaspimycin, optimizes ERS and proteotoxicity, leading to the suppression of uncontrollable rat sarcoma (RAS)-driven tumors [37].

Cellular adaptation to ERS is mediated by the UPR or ERSR, which works to restore the ER homeostasis. The cancer therapeutics targeted ERS-mediated apoptosis, through the ERSR, to kill tumor cells. For example, Osajin, a prenylated isoflavone, activates ERSR and the ER-resident cysteine protease, caspase-12, leading to caspase-3 activation and apoptosis in human nasopharyngeal carcinoma (NPC) [38]. The ERSR is mainly transduced by three ER-resident sensor proteins, (i) protein kinase RNA-like endoplasmic reticulum kinase (PERK), (ii) activating transcription factor 6 alpha (ATF6), and (iii) inositol-requiring enzyme 1 alpha (IRE1α) [39,40]. The luminal domains of these sensor proteins bind an ER chaperone, binding-immunoglobulin protein (BiP), that are separated at ERS conditions, which lead to the activation of ERSR [33]. The integrated signaling downstream of these three sensors, tightly controls life-or-death decisions in the cells exposed to ERS (Figure 2). The ERSR can induce pro-apoptotic signaling through the activation of DNA damage-inducible transcript 3, also known as the C/EBP homologous protein (CHOP), which subsequently leads to the initiation of apoptosis [41]. The CHOP can induce ERS-mediated apoptosis, through the down-regulation of anti-apoptotic Bcl-2 protein, and the upregulation of Bcl-2-associated X (Bax) protein [42,43]. Bcl-2 and Bax control a diversity of cellular responses to ERS, through the transport of Ca^2+^, in and out of the ER lumen. The Ca^2+^ released from ER through inositol 1,4,5-trisphosphate receptors (IP3Rs) and ryanodine receptors (RyRs) at ER and mitochondrial contact sites, can promote apoptosis, mainly via the release of cytochrome c from the mitochondria [44,45]. The cytochrome c then interacts with the apoptosis protease-activating factor-1 (Apaf-1), adenosine triphosphate (ATP), and pro-caspase-9 to form a supramolecular complex called the apoptosome, subsequently activating the caspase pathway through autocatalysis, and ultimately activating caspase-3, resulting in apoptosis (Figure 2) [46,47].

## 4. Chemotherapy Resistance: Role of Endoplasmic Reticulum Stress Response (ERSR)

### 4.1. Chemotherapy Resistance

Drug resistance refers to the reduction of the effectiveness of drugs, including antibiotics, antimalarial, antiviral, and chemotherapeutic agents, through different mechanisms during the treatment of various diseases [48]. Chemotherapy resistance is the tolerance of tumor cells to cancer treatments, such as chemotherapy, and targeted therapies. Chemotherapy resistance can be primary or acquired, which result in lowering drug accumulation and increasing drug efflux, or alterations of drug targets and signaling transduction molecules, leading to the repair of drug-induced DNA damage, subsequently blocking apoptosis, to kill tumor cells [49]. Primary or innate drug resistance occurs prior to any given treatment, whereas acquired resistance develops after the initial therapy [49]. Therapeutic resistance is a major barrier to the improvement of outcomes, for a variety of cancers [50].

The underlying mechanisms of cancer drug resistance are complex, and involve alteration in drug transports and targets, modification of signaling transduction molecules, or reactivation of antioxidant systems, leading to an impaired apoptosis.

#### Altered Membrane Transport

Drug transport across biological membranes is possible through passive trans-cellular transport and carrier-mediated transporters. Carrier-mediated transporters are of great interest to cancer researchers, as the transporters are responsible for the uptake, accumulation, distribution, and the efflux of drugs [51]. As examples, drug influx transporters including organic anion transporter (OAT), organic anion-transporting polypeptide (OATP), organic cation transporter (OCT), novel organic cation-transporter (OCTN), oligopeptide transporter (PEPT), and copper transporter (Ctr), are responsible for the uptake of most antineoplastic drugs [52,53,54]. The ATP-dependent multidrug transporters are in the class of carrier-mediated transporters, which are in the ubiquitous superfamily of ATP-binding cassette (ABC) proteins. ABC proteins, like multidrug resistance protein (MRP) and breast cancer resistance protein (BCRP) actively efflux chemotherapeutic agents out of the cell [55,56,57,58,59].

Alteration of the membrane transport system is one of the most important factors of cancer therapeutic resistance, in which the membrane proteins extrude chemotherapeutic agents, leading to a lower concentration of the drugs required to kill the target cells, and ultimately, a failure of chemotherapy [60]. As an example, the over-expression of ABC transporters on the plasma membrane can promote drug resistance in cancer cells [61]. ABC transporters have also been linked to resistance to infectious diseases, such as acquired immune deficiency syndrome (AIDS) and malaria [62,63]. MRP1 has been upregulated in different tumors and overexpressed in various cancer cells, during anticancer treatments [64,65]. MRP1 transporters were found to be up-regulated in colon cancer, which can facilitate the efflux of anticancer drugs out of cancer cells and decrease their therapeutic effects [66,67]. In addition to their physiological roles in host detoxification and pharmacokinetics, dysregulation of ABC transporters is associated with a variety of diseases.

### 4.2. Alteration of Drug Targets

Alteration of the molecular target significantly influences the effectiveness of cancer therapy, such as mutations or modifications of gene expression levels, leading to drug resistance. For example, active anticancer drugs, notably etoposide and doxorubicin-targeting DNA topoisomerase II (Topo II), play critical roles in DNA replication, transcription, and chromosome segregation [68,69,70,71]. Topoisomerase-targeting anticancer drugs act through (i) topoisomerase poisoning, which leads to a replication fork arrest and double-strand break formation, and (ii) topoisomerase inhibition, which leads to inhibition of the ATPase catalytic activity. The cancer cells can resist cancer therapy through various means. For example, some cancer cells have become resistant to Topo II-inhibiting drugs, through mutations in the Topo II gene, and some by means of increasing or decreasing its activities [72,73,74]. MRP1 confers resistance to Topo II-poisoning drugs (e.g., doxorubicin and etoposide), which is associated with a decreased cellular accumulation of cytotoxic drugs [75,76].

### 4.3. Activation of Antioxidant and Detoxification Systems

Chemotherapy-resistant cancer cells show a high expression of antioxidants and detoxifying systems. Most anticancer drugs require metabolic activation to achieve clinical efficacy. This means that the cancer cells can develop resistance through decreasing drug activation. For example, the anticancer drugs doxorubicin and oracin are inactivated by aldo-keto reductase (AKR) 1C3 [77]. Similar to oracin, doxorubicin goes through a metabolic detoxification by carbonyl reduction to the corresponding C13 alcohol metabolite [78]. The platinum-based cancer drug encounters resistance through the activation of detoxification systems by metallothionein and thiol glutathione [79]. Glutathione (GSH) and GSH-metabolism play an important role in cancer prevention and progression. GSH depletion has been shown to sensitize cells to compounds that produce oxidative cytolysis [80]. Elevation of Glutathione S-transferases (GST), which are also associated with resistance to apoptosis, can be stimulated by a variety of stimuli [81]. GSTs also play a role in the development of drug resistance through direct detoxification, by inhibition of the mitogen-activated protein kinase (MAPK) pathway [82].

### 4.4. Endoplasmic Reticulum Stress Response (ERSR) Is one of the Major Signaling Pathways of Chemotherapy Resistance

ER is an essential site of cellular homeostasis regulation, which can be hampered by various physiological and pathological conditions, resulting in ERS [83]. Recently, it has been demonstrated that cancer cells with constitutive or acquired resistance to chemotherapy are also resistant to ERS-triggered cell death [19]. The ERSR is a pathway of recent interest to many investigators, due to its role in adaptive survival signaling in cancer. The ERSR represents an adaptive mechanism that supports survival and chemo-resistance of tumor cells. The ERSR is regulated by ATF6, IRE1, and PERK in normal cells [84], while it is often deregulated and promotes tumorigenicity, as the depletion of tumor suppressors or activation of oncogenes, favors cells that survive during high protein synthesis and metabolic stress in cancer cells [85].

#### 4.4.1. Role of Protein Kinase RNA-Like ER Kinase (PERK) in Chemotherapy Resistance

ERSR or UPR is initiated in cancer cells, through three ER sensors ATF6, IRE1, and PERK [86,87]. ERS-mediated PERK-dependent signaling is critical for cell survival or drug resistance, following the initiation of an ERSR (Figure 3). For example, the survival rate has been dramatically reduced in PERK-deleted embryonic stem cells, following exposure to ERS-inducing agents [88]. The ERS-mediated activation PERK phosphorylates eukaryotic translation initiation factor-2α (eIF2α) and nuclear factor erythroid 2 (NFE2)-related factor 2 (Nrf2), leading to the attenuation of protein translation and an increase of genes controlling the redox homeostasis [87,89].

Most of the biological activities of PERK have been initiated through its downstream target translational initiation factor eIF2α kinase and subsequent translational upregulation of ATF4. A cell with compromised PERK/eIF2α/ATF4 signaling confers a survival advantage of tumor cells, under hypoxia [90]. Hypoxia is a unique feature of the tumor cells that contributes to cancer’s progression. To overcome hypoxic stress, tumor cells initiate hypoxia-inducing factor (HIF)-dependent transcriptional responses, through hypoxia-inducible factors HIF-1 and HIF-2, to adapt and survive [91,92]. It has been reported that PERK/eIF2α signaling plays a critical role in protecting chemotherapeutic resistant hypoxic cells, through induction of GSH synthesis, thus, reducing accumulation of ROS [93]. Combining hyperthermia with a low linear energy transfer (LET) radiation, could be the effective strategy to overcome hypoxia, to make cells sensitive to treatment, again [94,95].

PERK kinase and its downstream target, Nrf2, is the master transcriptional regulator of the cellular antioxidant and detoxifying systems, which are key mediators of chemotherapy resistance, in various tumor cells [96]. Nrf2 has been identified as a novel transcription substrate of PERK and is associated with Kelch-like ECH-associated protein 1 (Keap1), in unstressed cells, where activation of PERK via agents that trigger the ERSR is necessary and sufficient for dissociation of cytoplasmic Nrf2/Keap1 and a subsequent Nrf2 nuclear import [89,97]. A strong positive correlation between the Nrf2 levels and the resistance of three cancer cell lines has been shown for chemotherapeutic drugs, such as cisplatin, doxorubicin, and etoposide [98]. Nrf2 has been involved in the promotion of cell survival or chemotherapy resistance to cancer cells, following ERS [89]. Nrf2 is an attractive molecule as a therapeutic target in cancer. For example, ML385, a small molecule inhibitor of Nrf2, selectively alters therapeutic resistance in Keap1-deficient Non-small-cell lung carcinoma (NSCLC) tumors [99]. The ability of Nrf2 to activate cyto-protective detoxifying enzymes systems, including the cytochrome P450 (CYP) system, glutathione-S-transferase (GST), and uridine diphospho-glucuronosyltransferase (UGT), plays a crucial role in reducing ROS, leading to the prevention of apoptosis in cancer cells [100,101]. Nrf2 basically regulates the phase II detoxifying enzymes system, upon the exposure of cells to agents that promote the accumulation of ROS [102]. GSTs have been implicated in the detoxification of many chemotherapeutics. Elevated levels of GSTs have been associated with malignant transformation and drug resistance [103,104]. Overexpression of GST in cancer cells, enhances the detoxification of the anticancer drugs, leading to a reduction in the efficacy of cytotoxic damage to the cells [105]. Elevation of GST is also associated with resistance to apoptosis, through direct detoxification, by inhibition of the MAPK pathway [79,80]. Overexpression of GSTs ensures a high GSH concentration and increases GSH-transporters inside cancer cells, which are associated with high resistance to chemotherapeutic agents [106]. Several studies have suggested that a high GSH level is associated with an apoptotic resistance to the cancer cells [107,108]. The GSH metabolism has been reported to contribute to cancer cell resistance to anti-neoplastic treatments, either by inhibition of apoptosis, or through chemotherapeutic drug detoxification, by a GSH-conjugation [109]. The chemotherapeutic drugs are conjugated with GHS and efflux, through the GSH-pump of MRP transporters [110]. Nrf2-dependent proteins exert a pro-survival effect that can be through inducing HIF𝛼, increasing MRP or increasing the synthesis of GSH [111,112,113].

Multidrug resistance (MDR) is one of the main obstacles to treating cancer, as it can facilitate the efflux of anticancer drugs from cancer cells [64,65,114]. For example, PERK has been involved in the upregulation of MDR-related protein, MRP1, through the regulation of Nrf2 in the chemotherapy-resistant human colon adenocarcinoma cells line (HT29) [17]. In cancer cells, MDR is developed by upregulating antioxidant enzymes (hemeoxygenase 1, superoxide dismutase, catalase, etc.) that neutralize the ROS required for chemotherapy toxicity or by upregulating drug efflux pumps [101,115,116]. It has been reported that PERK-Nrf2 signaling protects de-differentiated cells from chemotherapy, by reducing ROS levels and increasing drug efflux, which suggests that targeting this pathway could sensitize drug-resistant cells to chemotherapy [97].

The ratio of anti-apoptotic Bcl-2 and apoptotic Bax proteins largely regulates cell death and survival [117,118,119]. It has been reported that Nrf2 exhibits protection against apoptosis, by increasing the levels of anti-apoptotic protein Bcl-2, in the hydrogen sulfide (H_2_S)-treated mice [120]. Elevated expression of Bcl-2 protein is associated with poor prognosis in many human cancers. It has been demonstrated that antioxidant exposure initiates Nrf2 stabilization, which leads to an increased Bcl-2/Bax ratio, subsequently reducing apoptosis and increasing cell survival [121]. The elevated expression of Bcl-2 protects lung cancer A549 cells from drug or radiation-induced DNA fragmentation and apoptosis [122,123].

It has been assumed that over-expression of Bcl-2 family members and the loss of wild-type p53 are mainly involved in resistance to apoptosis induced by chemotherapeutic drugs [124]. Bcl-2 constitutively controls p53-dependent apoptosis in HCT116 cellosaurus human colorectal cancer cells and A2780 ovarian cell line [125,126]. The tumor suppressor gene p53 plays an important role in chemotherapy resistance. Nrf2 regulates the basal expression of a direct inhibitor of p53, namely mouse double minute 2 homolog (Mdm2) [127]. The elevated expression of Nrf2 can cease ROS-based apoptotic signals, through the inhibition of p53, thus, enhancing cellular survival by leading to chemotherapy resistance [128]. It has also been reported that the p53/p21 complex regulates tumor cell invasion and apoptosis, by targeting Bcl-2 family proteins [129]. Nrf2 can inhibit the p53 target gene p21 by blocking interaction with Keap1 [130]. It has been found that Bcl-2 overexpression is associated with an increased resistance to cell death, after treatment with etoposide (DNA topoisomerase II inhibitor) [131]. The treatment of etoposide induces apoptosis, which is accompanied by the down-regulation of Bcl-2 expression, in small cell lung cancer (SCLC) and NSCLC cell lines [132].

The mammalian inhibitor of the apoptosis (IAP) gene family encoding proteins (XIAP, cIAP1 and cIAP2), contributes towards protecting the cells against a variety of apoptotic stimuli, resulting in drug resistance [133,134]. The cellular IAPs (cIAP1 and cIAP2) and the X-chromosome-linked IAP (XIAP) are highly expressed in many human malignancies [135,136,137,138,139]. It has been demonstrated that PERK activity can induce cellular inhibitor of apoptosis (cIAP1 and cIAP2) proteins, leading to the inhibition of ER stress-induced apoptotic program [140]. The ER stress inducing agents can activate the phosphatidylinositol 3-kinase (PI3K)–Akt signaling pathway, which is involved in the transcriptional induction of IAPs [141,142,143,144].

#### 4.4.2. Role of Inositol-Requiring Enzyme 1 Alpha (IRE1), Activating Transcription Factor 6 Alpha (ATF6), and Glucose-Regulated Protein 78 (GRP78) in Chemotherapy Resistance

The inositol-requiring enzyme 1 alpha (IRE1), activating transcription factor 6 alpha (ATF6), and glucose-regulated protein 78 (GRP78), play a critical role in chemotherapy resistance in cancers (Figure 4).

IRE1 could regulate cell survival induced by the ERS inducing agents, tunicamycin, and thapsigargin [145]. IRE1α, a subset of the ERSR, controls cyclin A1 expression and mediates proliferation, through tight control of XBP-1 splicing [146]. It has been established that XBP1 and HIF1α co-operatively regulate HIF1α targets, including vascular endothelial growth factor-A (VEGF-A), phosphoinositide-dependent kinase 1 (PDPK1), glucose transporter 1 (GLUT1), and DNA-damage-inducible transcript 4 (DDIT4), in triple-negative breast cancer (TNBC) [147].

Upon ERS, IRE1 recruits the tumor necrosis factor receptor (TNFR)-associated factor-2 (TRAF2) and activates apoptosis-signaling-kinase 1 (ASK1), leading to the activation of c-jun N-terminal protein kinase (JNK) [148,149]. ERSR can stimulate JNKs, which are important in controlling apoptosis in the process of cellular stress, thus, increasing the activator protein 1 (AP-1) transcriptional activity. It has been demonstrated that tamoxifen resistance is accompanied by an increased AP-1 DNA binding in MCF-7 cells [150].

The Raf/MEK/ERK pathway plays an important role in chemotherapeutic drug resistance, as activation of Raf induces resistance to doxorubicin and paclitaxel, in breast cancer cells [151,152]. The activation of extracellular signal-regulated kinase-1 and 2 (ERK1/2), is partially IRE1-dependent in azetidine-2 carboxylic acid-treated mouse embryonic fibroblasts (MEFs), in vitro [153]. Upon ER stress, the non-catalytic region of tyrosine kinase adaptor protein 1 (Nck1), dissociated from the ER membrane, increased activation of ERK1/2 and was correlated in vivo, with increased cell proliferation and reduced apoptosis [154].

ATF6 activation is essential for protecting melanoma cells against ERS-induced cell death [155]. Two Golgi resident enzymes, site-1 protease (S1P) and site-2 protease (S2P), are involved in the proteolytic cleavage of the full-length 90-kDa ATF6 [156,157]. It has been identified that Notch1 acts as a novel transcriptional target of ATF6, with a potential role in promoting an anti-apoptotic phenotype in irradiated glioblastoma [158]. The induction of ATF6α was required for cell survival and the expression of ER chaperones, upon ERS in the ATF6α knockout mouse embryonic fibroblasts (MEFs) [159,160]. The expression of the ER chaperone BiP protects the cells from ERS [161]. A novel role for ATF6α in chemo-resistance through the protein disulfide isomerase A5 (PDIA5) upon ER stress has been identified, in which genetic and pharmacological inhibition of the PDIA5/ATF6α, could restore sensitivity to the chemotherapy treatment [162]. The transcription factor ATF6α has been identified as a pivotal survival factor for quiescent squamous carcinoma cells [163]. Knockdown of ATF6α prolonged the survival of nude mice bearing dormant tumor cells through the ATF6α-Rheb-mammalian target of rapamycin (mTOR) pathway in a p38-dependent manner.

GRP78, also referred to as BiP, is a major molecular chaperone protein with anti-apoptotic properties, which plays a key role as a pro-survival component of ERSR [164,165,166]. The level of GRP78 is elevated in a variety of cancer cell lines and solid tumors associated with malignancy, and correlates with cellular drug resistance [167,168,169,170,171]. It has been demonstrated that under-expression of GRP78, induces apoptosis of colon cancer cells, in vitro, and suppresses tumor formation of colon cancer cells in a mouse xenograft model [172]. Enhanced pro-survival signaling resulting from ERSR has been found to promote resistance to chemotherapy, through upregulation of the canonical targets, such as GRP78 [173]. The induction of GRP78 could lead to cancer progression and drug resistance in neoplastic cells. On the other hand, overexpression of GRP78 could limit damage in normal tissues and organs exposed to ERS, due to its anti-apoptotic function [164]. Overexpression of GRP78 induced by celecoxib, partially suppresses the induction of CHOP and protects cancer cells from celecoxib-induced apoptosis in human GC cells [174]. It has been revealed that overexpression of GRP78, suppresses apoptosis induced by the Bcl-2-interacting killer (BIK) and NOXA, leading to activation of Bcl-2 and blocking of caspase-dependent apoptosis [175,176]. It has been reported that GRP78 protects cells from apoptosis induced by Topo II inhibitors (etoposide), by blocking the activation of caspase-7, both in vivo and in vitro [176,177]. The evidence suggests that mutation or altered expression of the Topo II genes are sufficient to confer resistance to topoisomerase inhibitors [178]. The under-expression of Topo II in tumor cells, due to mutations and posttranslational modification, leads to a reduction in the formation of the cleavable-complex in cancer cells, thus, preventing apoptosis of tumor cells [179,180]. Several studies have demonstrated that ERSR leads to a decrease in Topo II α protein and a concomitant resistance to chemotherapeutic agents that target this enzyme [168,181]. It has been reported that ERSR is essential and sufficient for reducing Topo IIα protein levels, thereby, increasing the resistance to topoisomerase-targeted drugs, like etoposide and doxorubicin [182]. The glucose-regulated proteins (GPRs) actively reduced Topo II expression in isolated nuclei from K12 cells, during a stress response, which is responsible for cellular resistance to etoposide [181,183].

The anticancer drugs 5-fluorouracil (5-FU) and doxorubicin become desensitized to renal cell carcinoma (RCC), due to the silencing of GRP78, by small interfering ribonucleic acid (siRNA) that induced G1 cell-cycle arrest and inhibited the growth of cells [184]. The specific inhibition of GRP78 expression suppressed G1/S transition-related cyclins (D1, E1, and E2) and cyclin-dependent kinase (CDK4 and CDK6) protein expression [184].

A study has demonstrated that GRP78 secreted by colon cancer cells facilitates cell proliferation via the PI3K/Akt signaling. It also showed that PI3K/Akt inhibition promoted ERS-induced apoptosis in a GRP78-dependent manner [185]. GRP78 contributes to the proliferation and tumorigenesis of human colorectal carcinoma, via the Akt and MAPK pathways [186]. An emerging study indicated that binding of activated alpha2-macroglobulin (α2-macroglobulin) to surface GRP78 activates p21-activated kinases (PAKs)-dependent signaling and promotes proliferation and metastasis in 1-LN prostate cancer cells [187]. The phosphorylation of GRP78, leads to the activation of Ras/MAPK and PI3-kinase downstream signaling, and recruitment of PAK-2, via the adaptor protein, NCK to the plasma membrane, thus, inhibiting the pro-apoptotic protein Bcl-2-associated death (Bad) [183].

## 5. Conclusions

Chemotherapy resistance is the tolerance of tumor cells to cancer treatments, which is a significant issue in the management of cancer. As such, chemotherapy resistance can be a major barrier for the physician in the treatment of cancer. Only a thorough understanding of the mechanisms of chemotherapy resistance will allow the physician to set the therapy as needed. While the development of new chemotherapies has proceeded quickly, chemotherapeutic agents that are effective against the advanced stages of cancer (such as invasion and metastasis) have not been discovered, due to a poor understanding of chemotherapy resistance. The ERSR has a vital role in the adaptive survival signaling that promotes resistance to chemotherapy in cancer cells. The chemotherapy resistance resulting from the ERSR is more or less regulated by three ER-resident sensor proteins, namely PERK, ATF6, and IRE1α. In these circumstances, ERSR could become a potential and effective target for the new development of chemotherapeutics. However, the detailed molecular mechanisms of the relationship between the ERSR and chemotherapy resistance remain poorly understand. In this review, we explained the molecular mechanism of ER stress-mediated apoptosis on cancer treatments, and also correlated the most remarkable pathways of the interplay between the ERSR and chemotherapy resistance.

## Figures and Tables

**Figure 1 cancers-11-00338-f001:**
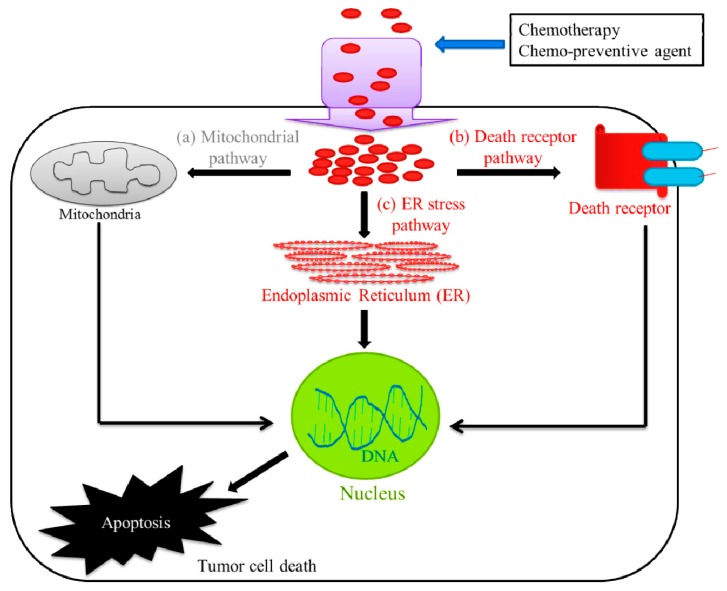
Pathways involved in tumor cell death. The chemotherapeutic treatment or chemotherapy mainly utilize three different pathways that lead to tumor cell death or apoptosis, which are (a) the extrinsic death receptor pathway, (b) the intrinsic mitochondrial pathway, and (c) the endoplasmic reticulum stress (ERS) pathway.

**Figure 2 cancers-11-00338-f002:**
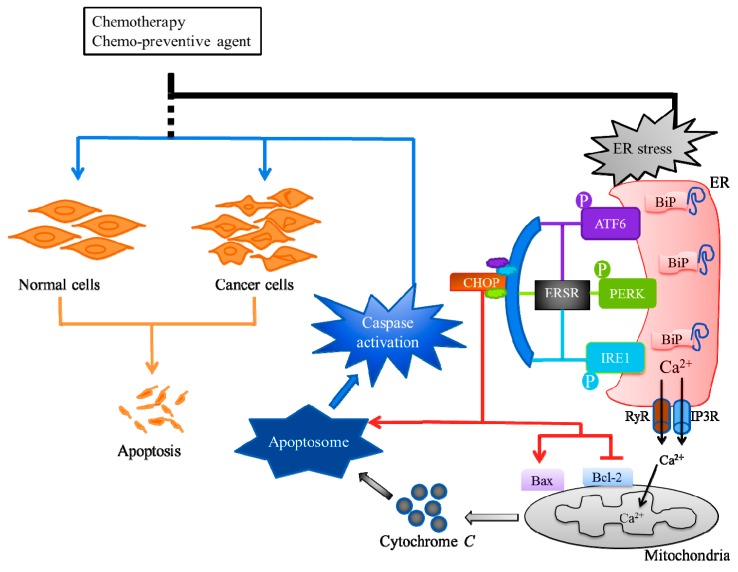
Endoplasmic reticulum stress (ERS)-mediated apoptosis and cancer therapy. Cancers can be treated with radiotherapy or chemotherapy, either individually or as a combination therapy. The chemotherapeutic exerts its anticancer activity by inducing ERS through endoplasmic reticulum stress response (ERSR) to kill tumor cells. The ERSR is mainly transduced by three ER-resident sensor proteins, protein kinase RNA-like endoplasmic reticulum kinase (PERK), activating the transcription factor 6 alpha (ATF6), and inositol requiring enzyme 1 alpha (IRE1). The integrated signaling downstream of these three sensors can induce pro-apoptotic signaling through the activation of C/EBP homologous protein (CHOP) that can downregulate B-cell lymphoma-2 (Bcl-2) protein and upregulate Bcl-2-associated X (Bax) protein. The Ca^2+^ released through inositol 1,4,5-trisphosphate receptors (IP3Rs) and ryanodine receptors (RyRs) at ER and mitochondrial contact sites can promote the release of mitochondria cytochrome c which interacts with apoptosis protease-activating factor-1 (Apaf-1), adenosine triphosphate (ATP) and procaspase-9 to form the apoptosome, and subsequently activates the caspase pathway resulting in apoptosis.

**Figure 3 cancers-11-00338-f003:**
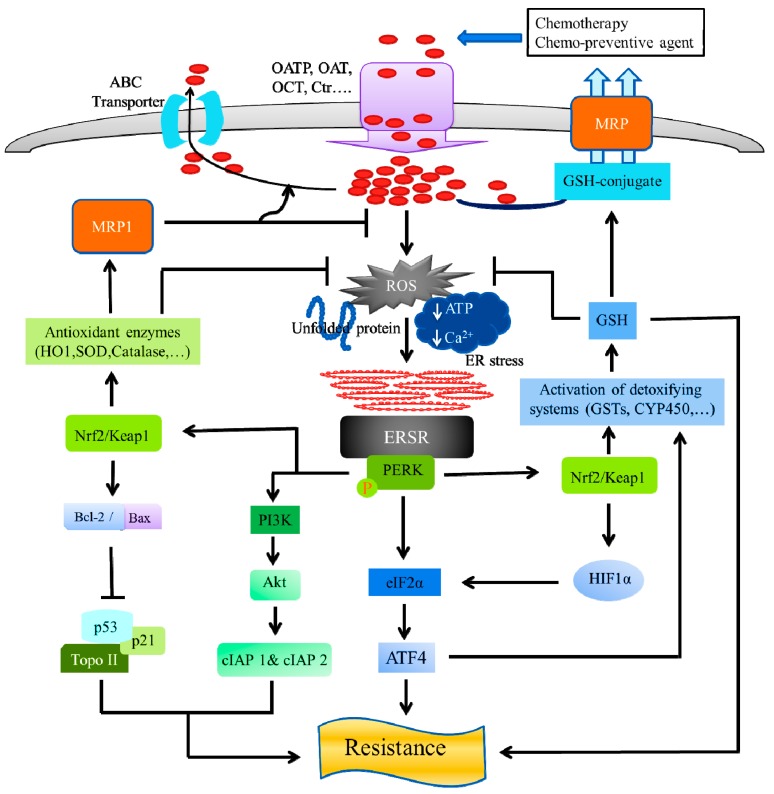
ER stress and chemo-resistance: Role of protein kinase R-like endoplasmic reticulum kinase (PERK). The anti-neoplastic drugs, possess ERS-mediated apoptosis and exert their anticancer effect by entering the tumor cells, using influx transporters, including an organic anion transporter (OAT), organic anion-transporting polypeptide (OATP), organic cation transporter (OCT), and a copper transporter (Ctr). On the other hand, ATP-dependent multidrug transporters of the carrier-mediated, ubiquitous superfamily of ATP-binding cassette (ABC) proteins actively efflux chemotherapeutic agents out of the cell. ABC proteins include the multidrug resistance protein (MDR), which confer resistance to many anticancer drugs in cancer cells. The ERS-mediated activated PERK, phosphorylates eukaryotic translation initiation factor-2α (eIF2α) and nuclear factor erythroid 2 (NFE2)-related factor 2 (Nrf2). Activation of PERK via agents that trigger the ERSR, promotes a dissociation of the cytoplasmic Nrf2/Keap1, and the Nrf2 possess its cyto-protective activity by activating detoxifying enzyme systems, including the cytochrome P450 (CYP) system, glutathione-S-transferases (GSTs), thus, reducing ROS, leading to the prevention of apoptosis in cancer cells. The GSTs ensure a high GSH concentration and GSH-transporters inside cancer cells, which are associated with a high resistance to chemotherapeutic agents. The GSH exhibits cancer cell resistance to anti-neoplastic treatments, either by inhibition of apoptosis or chemotherapeutic drug detoxification, by the GSH-drug conjugation, followed by efflux through the GSH-pump of MRP transporters. The activated PERK-Nrf2 signaling, upregulates the multidrug resistance protein 1 (MRP1), by the activation of antioxidant enzymes (HO-1, SOD, catalase, etc.) that neutralize the reactive oxygen species (ROS) and increases the drug efflux, leading to a reduction in the ROS levels. Nrf2 stabilization by the exposure of antioxidant enzymes, leads to an increased Bcl-2/Bax ratio, subsequently reducing apoptosis and increasing cell survival, by regulation of Topo II, p53, and its target gene p21. The PERK activity can induce cellular inhibitor of apoptosis (cIAP1 and cIAP2) proteins, by the activation of phosphatidylinositol 3-kinase (PI3K)-Akt signaling pathway, leading to chemotherapy resistance.

**Figure 4 cancers-11-00338-f004:**
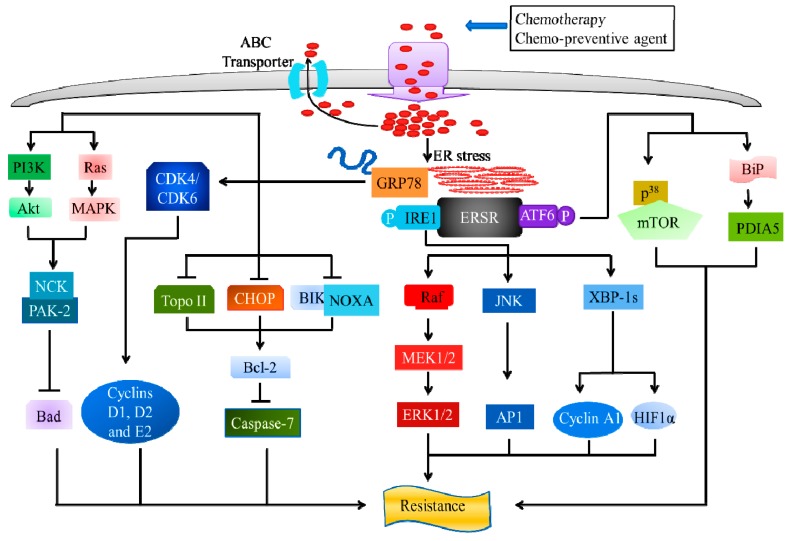
ER stress and chemo-resistance: Role of inositol-requiring enzyme 1 alpha (IRE1), activating transcription factor 6 alpha (ATF6), and glucose-regulated protein 78 (GRP78). Upon ERS, IRE splices and activates the X-box-binding protein 1 (XBP1) that controls cycline A1 expression and regulates HIF1α targets. The activation of IRE1 leads to the activation of c-Jun N-terminal protein kinase (JNK), followed by inducing AP-1 transcriptional activity. The Raf/MEK/ERK pathway is also involved in IRE1-mediated chemo-resistance. ATF6α possesses chemo-resistance through the regulation of ER chaperone BiP protein, mediated by the activation of disulfide isomerase A5 (PDIA5) and p38-dependent ATF6α-mTOR pathway, upon ER stress. GRP78 exhibits its chemo-resistance properties, upon ERS, through the suppression of C/EBP homologous protein (CHOP), Bcl-2-interacting killer (BIK), phorbol-12-myristate-13-acetate-induced protein 1 (PMAIP1) or NOXA, Topo II, and induction of Bcl-2, leading to the activation of caspase-7 dependent apoptosis. GRP78 suppresses G1/S transition-related cyclins (D1, E1, and E2) and cyclin-dependent kinase (CDK4 and CDK6) protein expression. Additionally, the phosphorylation of GRP78 leads to the activation of Ras/MAPK and PI3-kinase downstream signaling, and the recruitment of p21 activated kinases 2 (PAK-2) via NCK, thus, inhibiting the pro-apoptotic protein Bcl-2-associated death (Bad).

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
