# Peer review of "Chemotherapy Resistance Explained through Endoplasmic Reticulum Stress-Dependent Signaling"

_cancers, 2019, doi:10.3390/cancers11030338_

Round 1
Reviewer 1 Report
The Endoplasmic reticulum stress response in relation to resistance to chemotherapy is discussed, including the ERS induced apoptosis. Indeed, the ERSR can be a target of chemotherapy as well. In the review molecular mechanism of ER stress‑mediated apoptosis on cancer treatments are described. Correlations with pathways of the interplay between the ERSR and chemotherapy resistance are presented. However, no clue is given how to overcome this resistance. Is nothing known? May be a short discussion can be given on strategies to overcome ERSR chemo resistance.
The authors come from Korea and then the first sentence of the introduction is on the occurrence of cancer in the US! Why not about Korea, Asia or better world wide?
In line 46 (and also in para 4.1) the authors state that “Chemotherapy, radiotherapy and targeted therapies encounter resistance through different mechanisms”. Do the authors mean here acquired resistance to treatment? I do not think this is true for radiation although some cells have better DNA repair and are less radiation sensitive but acquired resistance to radiation does not occur, I think. Cells do not develop a resistance to radiation. In the end the normal tissue late radiation toxicity is a problem which is why not can be continued with radiotherapy.
In para 2 different therapies are mentioned including γ-irradiation. This should be changed into radiotherapy as treatment of cancer with non invasive radiotherapy in the Western world is mostly carried out with Linacs (electrons) or photons from X-ray. The main cause of radiation induced cell reproductive death in solid cancers after clinically applied doses is mitotic cell death. For the bone marrow apoptosis is also an important cause of cell death. So may be the radiation induced cell death can be shortly discussed differently.
In para 4.4.1 hypoxia is mentioned and then the PERK/eIF2α signaling is mentioned to play role in protecting chemotherapy resistant hypoxic cells. Unfortunately, nothing is said to overcome hypoxia to make cells sensitive for treatment again. May be hyperthermia can be mentioned here as a strategy to overcome hypoxia (Elming et al. Cancers 2019, 11(1), 60; https://doi.org/10.3390/cancers11010060).
This the first sentence of the conclusion is: “Chemotherapy resistance is the tolerance of tumor cells to cancer treatments including chemotherapy, radiotherapy and targeted therapies, and operates through many different mechanisms”. I think this sentence is not correct and need to be changed.
The manuscript end with: “The present review aims to explain….”. This is wrong semantics a review does not explain, people do. May be, it is better to write: “In the review it is explained” or “In the review explanations on…. are presented….”. Something like this.
Author Response
The Endoplasmic reticulum stress response in relation to resistance to chemotherapy is discussed, including the ERS induced apoptosis. Indeed, the ERSR can be a target of chemotherapy as well. In the review molecular mechanism of ER stress‑mediated apoptosis on cancer treatments are described. Correlations with pathways of the interplay between the ERSR and chemotherapy resistance are presented. However, no clue is given how to overcome this resistance. Is nothing known? May be a short discussion can be given on strategies to overcome ERSR chemo resistance.
Response: Thank you for your suggestion. We have given some short description or example during explanation of role of ER-residence protein in chemotherapy resistance to overcome chemotherapy resistance.
Line 288-290: Combining hyperthermia with low linear energy transfer (LET) radiation could be the effective strategy to overcome hypoxia to make cells sensitive for treatment again….
Line 300-302: Nrf2 is an attractive molecule as a therapeutic target in cancer. For example, ML385, a small molecule inhibitor of Nrf2 selectively alters therapeutic resistance in Keap1-deficient Non-small-cell lung carcinoma (NSCLC) tumors….
The authors come from Korea and then the first sentence of the introduction is on the occurrence of cancer in the US! Why not about Korea, Asia or better world wide?
Response: We have revised
In line 46 (and also in para 4.1) the authors state that “Chemotherapy, radiotherapy and targeted therapies encounter resistance through different mechanisms”. Do the authors mean here acquired resistance to treatment? I do not think this is true for radiation although some cells have better DNA repair and are less radiation sensitive but acquired resistance to radiation does not occur, I think. Cells do not develop a resistance to radiation. In the end the normal tissue late radiation toxicity is a problem which is why not can be continued with radiotherapy.
Response: We have revised to clarify the meaning of the sentence.
In para 2 different therapies are mentioned including γ-irradiation. This should be changed into radiotherapy as treatment of cancer with non invasive radiotherapy in the Western world is mostly carried out with Linacs (electrons) or photons from X-ray. The main cause of radiation induced cell reproductive death in solid cancers after clinically applied doses is mitotic cell death. For the bone marrow apoptosis is also an important cause of cell death. So may be the radiation induced cell death can be shortly discussed differently.
Response: Thank you for your suggestion.
In para 4.4.1 hypoxia is mentioned and then the PERK/eIF2α signaling is mentioned to play role in protecting chemotherapy resistant hypoxic cells. Unfortunately, nothing is said to overcome hypoxia to make cells sensitive for treatment again. May be hyperthermia can be mentioned here as a strategy to overcome hypoxia (Elming et al. Cancers 2019, 11(1), 60; https://doi.org/10.3390/cancers11010060).
Response: We have revised according to reviewer suggestion.
This the first sentence of the conclusion is: “Chemotherapy resistance is the tolerance of tumor cells to cancer treatments including chemotherapy, radiotherapy and targeted therapies, and operates through many different mechanisms”. I think this sentence is not correct and need to be changed.
Response: We have revised the sentence and conclusion too.
The manuscript end with: “The present review aims to explain….”. This is wrong semantics a review does not explain, people do. May be, it is better to write: “In the review it is explained” or “In the review explanations on…. are presented….”. Something like this.
Response: We have revised
Reviewer 2 Report
Dear Lydia Thompson, B.Sc.
Cancers Editorial Office
I completed the revision of manuscript titled “Chemotherapeutic Resistance Explained Through Endoplasmic Reticulum Stress-Dependent Signaling”.
Overall, I consider this manuscript very interesting. However, I think that the manuscript require a minor revision before to be considered suitable for publication on Cancer Journal.
Minor point:
Abstract: lane 14, please remove “of”
Introduction: lane 50-55. This part has been explained in the abstract. I suggest to rewrite this paragraph to remove all repetitions.
Chapter 3: lane 139. Please remove the symbol “@”
Chapter 4.1.1: lane 221. Lane 221-224. The phrase:
“Alteration of the membrane transport system is one of the most important forms of cancer therapeutic resistance, in which the membrane proteins extrude chemotherapeutic agents, leading to a lower concentration of the drugs required to kill target cells”
it is very similar to the next sentence
“The efflux of drugs out of cancer cells through a variety of membrane transporters is the main cellular mechanism of the transport-based cellular drug resistance, leading to decreased intracellular accumulation of anticancer drugs and chemotherapy failure”
I suggest to rewrite this paragraph, or eliminate one of two phrases.
Sincerely
Author Response
I completed the revision of manuscript titled “Chemotherapeutic Resistance Explained Through Endoplasmic Reticulum Stress-Dependent Signaling”.
Overall, I consider this manuscript very interesting. However, I think that the manuscript require a minor revision before to be considered suitable for publication on Cancer Journal.
Minor point:
Abstract: lane 14, please remove “of”
Response: We have revised.
Introduction: lane 50-55. This part has been explained in the abstract. I suggest to rewrite this paragraph to remove all repetitions.
Response: We have rewrite.
Chapter 3: lane 139. Please remove the symbol “@”
Response: We have removed and add α
Chapter 4.1.1: lane 221. Lane 221-224. The phrase:
“Alteration of the membrane transport system is one of the most important forms of cancer therapeutic resistance, in which the membrane proteins extrude chemotherapeutic agents, leading to a lower concentration of the drugs required to kill target cells”
it is very similar to the next sentence
“The efflux of drugs out of cancer cells through a variety of membrane transporters is the main cellular mechanism of the transport-based cellular drug resistance, leading to decreased intracellular accumulation of anticancer drugs and chemotherapy failure”
I suggest to rewrite this paragraph, or eliminate one of two phrases.
Response: We have revised.
Reviewer 3 Report
In this review “Chemotherapeutic resistance explained through endoplasmatic reticulum stress-dependent signaling”, the authors collect the main factors involved in UPR and relate them to the resistance of tumors to different therapies, considering as well the utility of targeting apoptosis as potential therapy.
Please find the list of comments and suggestions in the text below.
During all the review, the authors write several times the same concept in different sentences just using different words. I would suggest the authors to merge all the information of these several sentences, and try to explain in a simpler way the certain concept, in order to do easier the flow of the text.
In addition, the authors do not use much linkers during the review which would render the text nicer and it would make easier the understanding of the different concepts given during the text.
1. Introduction
Line 38-43: I would try to merge the information of these sentences since they say the same.
In general, I would say that the first paragraph of the introduction says the same just been repeated in different words. Please, try to rearrange the text in order to improve the efficacy of the message that the authors want to give.
I would suggest to always use the same way of writing words such as “chemo-therapy” or “chemotherapy”.
3. Endoplasmic Reticulum Stress-mediated Apoptosis and Cancer Therapy
Line 134: “Cancer therapeutics target….”
Line 138: I would add i) ii) and iii)
Line 139: the alpha symbol of IRE1 is not well added.
Line 141: “lead to ERSR being triggered” I would suggest “lead to activation of ERSR”.
Line 152: …subsequently activating….
Line 153: …ultimately activating…
As authors represent in Figure 2, the use of chemotherapy targeting the ERS leading to apoptosis of both cancer cells and normal cells. Is there any way of avoiding the activation of apoptosis in normal cells since, as also said by the authors, is one of the main problems in chemotherapy? Can the authors introduce a small discussion about it since, in my opinion, it is an important and interesting point.
4.1. Chemotherapy Resistance
I would merge some sentences because there are 3 sentences saying the same in different ways. Maybe try, as indicated in the text (red, Lines 196-197): “Chemotherapy resistance is the tolerance of tumor cells to cancer treatments such as chemotherapy, radiotherapy and targeted therapies. ”
4.1.1. Altered Membrane Transport
I would merge the two paragraphs from line 219 to line 233 since they give the same information, just with different examples. So, sentences like “…ABC transporters can promote chemotherapy resistance in cancer cells…” are repeated and, in my opinion, can be combined.
4.2. Alteration of Drug Targets
I would just add some explanation on the last sentence where the authors talk about MRP1, just adding why it confers resistance to those drugs.
4.4. Endoplasmic Reticulum Stress Response (ERSR) is one of the Major Signaling Pathways of Chemotherapy Resistance
In the first piece of introduction (line 261-275), there is need of connectors between sentences in order to link and do the text smoother.
4.4.1. Role of Protein Kinase RNA-like ER Kinase (PERK) in Chemotherapy Resistance
Underlined in yellow (line 296-301): I would move this text to the beginning of the paragraph in order to give an introductory link between PERK and Nrf2. Following, there will be the examples/explanations of Nrf2, and afterwards the reasons why Nrf2 is an attractive molecule as therapeutic drug.
Line 326-328 (underlined in green): Please double check sentence, is not really clear.
In general, there is a need of connecting better all the different ideas that the authors are collecting in order to do the lecture smoother and clearer. A better way of linking sentences/ideas is also important.
In line with this, Figure 3 is a bit confusing: there is a lot of information and it is no very clear as a figure. I would suggest the authors to find a better way of representing the scheme in order to facilitate the visual understanding of the concepts within the figure. I would also try to refer more to the figure 3 while explaining the concepts in the text.
In the scheme of Figure 3, there are typos mistakes: IPA instead of IAP1 and 2.
4.4.2. Role of Inositol Requiring Enzyme 1 Alpha (IRE1), Activating Transcription Factor 6 Alpha (ATF6) and Glucose-regulated Protein 78 (GRP78) in Chemotherapy Resistance
Instead of Figure 3 is Figure 4.
5. Conclusion
In my opinion, there is a lack of actual conclusion, summarizing the major points of the review and extending the reasons why the “ERSR could become a potential and effective target for the new development of chemotherapeutics”. Therefore, I would ask the authors to improve the conclusions.

Author Response
In this review “Chemotherapeutic resistance explained through endoplasmatic reticulum stress-dependent signaling”, the authors collect the main factors involved in UPR and relate them to the resistance of tumors to different therapies, considering as well the utility of targeting apoptosis as potential therapy.
Please find the list of comments and suggestions in the text below.
During all the review, the authors write several times the same concept in different sentences just using different words. I would suggest the authors to merge all the information of these several sentences, and try to explain in a simpler way the certain concept, in order to do easier the flow of the text.
In addition, the authors do not use much linkers during the review which would render the text nicer and it would make easier the understanding of the different concepts given during the text.
1. Introduction
Line 38-43: I would try to merge the information of these sentences since they say the same.
In general, I would say that the first paragraph of the introduction says the same just been repeated in different words. Please, try to rearrange the text in order to improve the efficacy of the message that the authors want to give.
I would suggest to always use the same way of writing words such as “chemo-therapy” or “chemotherapy”.
Response: We have merged.
3. Endoplasmic Reticulum Stress-mediated Apoptosis and Cancer Therapy
Line 134: “Cancer therapeutics target….”
Response: We have revised.
Line 138: I would add i) ii) and iii)
Response: We have revised.
Line 139: the alpha symbol of IRE1 is not well added.
Response: We have added alpha symbol.
Line 141: “lead to ERSR being triggered” I would suggest “lead to activation of ERSR”.
Response: We have revised.
Line 152: …subsequently activating….
Response: We have revised.
Line 153: …ultimately activating…
Response: We have revised.
As authors represent in Figure 2, the use of chemotherapy targeting the ERS leading to apoptosis of both cancer cells and normal cells. Is there any way of avoiding the activation of apoptosis in normal cells since, as also said by the authors, is one of the main problems in chemotherapy? Can the authors introduce a small discussion about it since, in my opinion, it is an important and interesting point.
Response: Thank you very much for your suggestion.
4.1. Chemotherapy Resistance
I would merge some sentences because there are 3 sentences saying the same in different ways. Maybe try, as indicated in the text (red, Lines 196-197): “Chemotherapy resistance is the tolerance of tumor cells to cancer treatments such as chemotherapy, radiotherapy and targeted therapies. ”
Response: We have revised.
4.1.1. Altered Membrane Transport
I would merge the two paragraphs from line 219 to line 233 since they give the same information, just with different examples. So, sentences like “…ABC transporters can promote chemotherapy resistance in cancer cells…” are repeated and, in my opinion, can be combined.
Response: We have revised.
4.2. Alteration of Drug Targets
I would just add some explanation on the last sentence where the authors talk about MRP1, just adding why it confers resistance to those drugs.
Response: We have revised.
4.4. Endoplasmic Reticulum Stress Response (ERSR) is one of the Major Signaling Pathways of Chemotherapy Resistance
In the first piece of introduction (line 261-275), there is need of connectors between sentences in order to link and do the text smoother.
Response: We have revised.
4.4.1. Role of Protein Kinase RNA-like ER Kinase (PERK) in Chemotherapy Resistance
Underlined in yellow (line 296-301): I would move this text to the beginning of the paragraph in order to give an introductory link between PERK and Nrf2. Following, there will be the examples/explanations of Nrf2, and afterwards the reasons why Nrf2 is an attractive molecule as therapeutic drug.
Line 326-328 (underlined in green): Please double check sentence, is not really clear.
Response: Anticancer drugs initiate ERS-mediated apoptosis to kill cancer cell through induction of ROS production. This ROS become neutralize by the some antioxidant enzymes, lead to decrease cancer cell apoptosis, resulting in the development of Multidrug resistance (MDR). Also, these antioxidant enzymes activate multidrug resistance protein (MRP) which increases drug efflux from inside to outside the cells.
In general, there is a need of connecting better all the different ideas that the authors are collecting in order to do the lecture smoother and clearer. A better way of linking sentences/ideas is also important.
In line with this, Figure 3 is a bit confusing: there is a lot of information and it is no very clear as a figure. I would suggest the authors to find a better way of representing the scheme in order to facilitate the visual understanding of the concepts within the figure. I would also try to refer more to the figure 3 while explaining the concepts in the text.
In the scheme of Figure 3, there are typos mistakes: IPA instead of IAP1 and 2.
Response: We have revised.
4.4.2. Role of Inositol Requiring Enzyme 1 Alpha (IRE1), Activating Transcription Factor 6 Alpha (ATF6) and Glucose-regulated Protein 78 (GRP78) in Chemotherapy Resistance
Instead of Figure 3 is Figure 4.
Response: We have revised.
5. Conclusion
In my opinion, there is a lack of actual conclusion, summarizing the major points of the review and extending the reasons why the “ERSR could become a potential and effective target for the new development of chemotherapeutics”. Therefore, I would ask the authors to improve the conclusions.
Response: We have revised.